# Impact of Chronic Nitrate and Citrulline Malate Supplementation on Performance and Recovery in Spanish Professional Female Soccer Players: A Randomized Controlled Trial

**DOI:** 10.3390/nu17142381

**Published:** 2025-07-21

**Authors:** Marta Ramírez-Munera, Raúl Arcusa, Francisco Javier López-Román, Vicente Ávila-Gandía, Silvia Pérez-Piñero, Juan Carlos Muñoz-Carrillo, Antonio Jesús Luque-Rubia, Javier Marhuenda

**Affiliations:** 1Faculty of Pharmacy and Nutrition, Universidad Católica de Murcia (UCAM), Campus de los Jerónimos, Guadalupe, 30107 Murcia, Spain; marta.ramirez.nutricionista@gmail.com (M.R.-M.); jmarhuenda@ucam.edu (J.M.); 2Faculty of Medicine, Universidad Católica de Murcia (UCAM), Campus de los Jerónimos, Guadalupe, 30107 Murcia, Spain; jlroman@ucam.edu (F.J.L.-R.); vavila@ucam.edu (V.Á.-G.); sperez2@ucam.edu (S.P.-P.); jcmunoz@ucam.edu (J.C.M.-C.); ajluque@ucam.edu (A.J.L.-R.); 3Primary Care Research Group, Biomedical Research Institute of Murcia (IMIB-Arrixaca), 30120 Murcia, Spain

**Keywords:** ergogenic aids, nitric oxide booster, performance, recovery, female soccer

## Abstract

**Background**: Pre-season training is critical for developing tolerance to high physical demands in professional soccer, and nitric oxide (NO) precursors such as dietary nitrate (NO_3_^−^) and citrulline malate (CM) can support performance and recovery during this demanding phase. This study aimed to examine the effects of a four-week supplementation protocol combining 500 mg of NO_3_^−^ from amaranth extract and 8 g of CM (NIT + CM) on external training load and post-match recovery in professional female soccer players during pre-season. **Methods**: A randomized, double-blind, placebo-controlled trial was conducted with 34 female soccer players who received either the NIT + CM product or a placebo for four weeks during pre-season. Global positioning system (GPS)-derived external load was recorded throughout the intervention. Performance tests—a countermovement jump (CMJ) test and the Wingate anaerobic test (WAnT)—and blood sampling for plasma NO_3_^−^ and nitrite (NO_2_^−^) concentrations were conducted at baseline and the day after a competitive match. **Results**: The supplementation with NIT + CM increased maximal speed (Vmax) throughout training and match play. During post-match testing, the NIT + CM group exhibited a significantly smaller decline in mean (P_mean_) and minimum (P_min_) power during the WAnT, along with reduced power loss in both the first (0–15 s) and second (15–30 s) intervals. Plasma NO_3_^−^ concentrations significantly increased from baseline in the NIT + CM group and remained elevated 24 h after the final dose, confirming sustained systemic exposure. **Conclusions**: Chronic NIT + CM supplementation may enhance Vmax and help preserve anaerobic performance the day after a match. These effects could reflect improved tolerance to high training loads and sustained NO_3_^−^ availability during recovery.

## 1. Introduction

Soccer is an intermittent sport involving repeated high-intensity efforts interspersed with low-intensity activities [1,2]. Recent developments in women’s soccer, including its increased visibility and professionalization, have led to greater physical demands on players [3,4,5]. Consequently, the pre-season preparation period has become essential in developing the fitness required to sustain high training loads and match intensities throughout the competitive season and during tournaments [6,7,8].

Athletes, including female soccer players, frequently use dietary supplements to support training adaptations and competitive performance, with NO precursors ranking among the most widely marketed and accessible supplement groups [9]. NO is a signaling molecule involved in numerous physiological functions, including vasodilatation, mitochondrial respiration and biogenesis, angiogenesis, muscle glucose uptake, and sarcoplasmic reticulum calcium handling [10,11,12]. Through these mechanisms, NO has been linked to enhanced muscle efficiency [13,14], reduced oxygen demand during exercise, and improved neuromuscular recovery [15], making it a potential target for performance optimization.

The human body generates NO through two complementary pathways [16]. The first is the citrulline-arginine-NO pathway, in which NO is synthesized endogenously from L-arginine and oxygen through nitric oxide synthase (NOS) enzymes, producing L-citrulline as a byproduct [17]. This process is commonly referred to as the NOS-dependent pathway.

The second pathway is partially exogenous, as it relies on NO_3_^−^ and NO_2_^−^ obtained from both endogenous NO oxidation [18] and dietary sources rich in NO_3_^−^ [16] such as spinach, beets, and arugula [19]. A portion of the ingested NO_3_^−^ accumulates in saliva and is then reduced to NO_2_^−^ by bacteria in the oral cavity [20,21,22]. Subsequent ingestion of this NO_2_^−^ results in the reduction of a portion of it to NO within the acidic environment of the stomach [23], with the remainder entering the systemic circulation. This NO_2_^−^ is rapidly distributed in blood and other tissues, including peripheral vasculature and skeletal muscle [18], and can readily undergo a one-electron reduction to yield NO [9]. This reaction is enhanced under conditions of hypoxia [24] and acidity [25] such as those observed in skeletal muscle during contraction [26]. This route is known as the NO_3_^−^ -NO_2_^−^ -NO pathway or the NOS-independent pathway.

Given these pathways, it has been suggested that combining different NO precursors, such as dietary NO_3_^−^ and citrulline, may further enhance its bioavailability [27]. L-citrulline could support the NOS-dependent pathway by increasing L-arginine availability [28,29], while dietary NO_3_^−^ could contribute to the NOS-independent route by providing additional substrate for NO synthesis [18,30].

In this context, L-citrulline is commonly supplemented in the form of CM [31], a compound that consists of L-citrulline in combination with malic acid, an intermediate of the tricarboxylic acid cycle [32,33]. In addition to supporting NO synthesis, CM may contribute to enhanced ATP production and improved ammonia clearance, which could be particularly beneficial during repeated high-intensity efforts [34,35]. These mechanisms are particularly relevant in team sports such as soccer, which involve frequent explosive actions interspersed with short recovery periods [36] and constant alternation between anaerobic [37,38] and aerobic [39,40] energy systems.

Dietary NO_3_^−^ supplementation has been associated with performance improvements in endurance, muscular endurance, and high-intensity power output, as reported in a recent umbrella review summarizing 20 systematic reviews and meta-analyses [41].

These effects are thought to be mediated by NO–related physiological adaptations [10,11,12,13,14]. However, methodological variability and limited representation of female athletes highlight the need for further research in applied sport contexts [41,42,43,44].

In recent years, it has been suggested that type II muscle fibers—primarily recruited during explosive and high-intensity efforts [26]—may serve as NO_3_^−^ reservoirs, promoting localized NO production under hypoxic and acidic conditions and contributing to intramuscular buffering [45,46]. These observations support the hypothesis that skeletal muscle functions as a physiological NO_3_^−^ reservoir sensitive to dietary intake. Animal studies have shown that NO_3_^−^ depletion reduces intramuscular stores, while reintroduction leads to rapid recovery and even supraphysiological accumulation, suggesting a potential adaptive mechanism that may reinforce the rationale for chronic supplementation strategies [47]. Regarding the mode of administration, beetroot juice has been the most extensively studied source of inorganic NO_3_^−^ [48]. However, alternative dietary sources, such as amaranth, have gained attention in recent years as potential alternatives, although research on them remains limited [49,50,51].

Due to the physiological complementarity of CM and dietary NO_3_^−^, their combined use may offer a practical strategy to support performance and recovery by targeting distinct NO synthesis pathways. Although several studies have examined their effects separately [31,33,35,41,52,53], to our knowledge, no study has investigated their co-supplementation. This is the first study to evaluate the combined effects of CM and dietary NO_3_^−^ under a chronic supplementation protocol in professional soccer players. Exploring this strategy during pre-season may provide valuable insight into approaches aimed at attenuating fatigue, sustaining performance, and optimizing recovery in applied sport-specific settings.

The primary aim of this study is to examine whether chronic supplementation with 500 mg of NO_3_^−^ from amaranth extract and 8 g of CM over four weeks during pre-season influences external load and post-match recovery in professional female soccer players. We hypothesize that combined supplementation with CM and NO_3_^−^ may help attenuate performance decline and support recovery during pre-season.

## 2. Materials and Methods

### 2.1. Study Design

This study was a randomized, double-blind, team-stratified, and placebo-controlled trial designed to evaluate the effects of a four-week supplementation regimen during pre-season with NO_3_^−^ and CM (NIT + CM) on GPS-derived performance metrics and recovery one day after a match in professional female soccer players. On the first day of pre-season, CMJ, WAnT, and blood sampling were conducted. Three days later, supplementation began. GPS data were recorded daily during training sessions and matches over the 30-day period. On the final day of supplementation, both teams played a match, and all tests from the first day were repeated the following day. Prior to starting the intervention, the protocol was approved by the Institutional Review Committee of the Catholic University San Antonio of Murcia (UCAM) (code CE052204), following the Standards of Good Clinical Practice and was conducted according to the Declaration of Helsinki. The clinical trial was registered at www.clinicaltrials.gov (accessed on 9 August 2022) (identifier NCT05525871). Current European legislation on the protection of personal data (Regulation (EU)2016/679) was complied with.

### 2.2. Participants

Research was conducted on a sample of 34 professional female players (age: 23.06 ± 4.29 years) from two soccer teams in the first and second Spanish professional women’s football leagues.

The inclusion criteria for participant recruitment were as follows: (a) being a healthy subject with medical authorization for the practice of federated sport, and (b) belonging to a team of the first or second division of the Spanish Women’s League. The exclusion criteria for the study were as follows: (a) changing teams during pre-season, and (b) a history of drug, alcohol, or other substance abuse or other factors limiting the ability to cooperate during the study. All players were previously informed of the objectives and method of the research, signing informed consent forms before starting the research; in the case of minors, it was their parents who signed the consent. An outside researcher carried out the randomization using a computed generator (Epidat v4.1 Epi-dat, Spain), assigning the subjects to one of the groups (NIT + CM or PLA). Each product was assigned a unique code, ensuring there were as many codes as participants. A master sheet was prepared, linking each participant’s code with the corresponding product code. This allocation sheet was signed and dated to confirm the randomization process. The group assignments remained confidential and were disclosed only at the end of the study, identifying which codes corresponded to the NIT + CM and which to the PLA.

Player codes and group assignments were provided to the manufacturing company. Based on these codes, the company prepared individual bags containing the precise number of doses required for each player throughout the study. Each bag was labeled with the corresponding player code, ensuring that the correct product was assigned according to the randomization process. The company managed the separation of active supplement and placebo, maintaining blinding throughout the study.

### 2.3. Supplementation

The NIT + CM and PLA developed by Nutripeople (Reel & Innovation S.L., Murcia, Spain) was supplied in flexible 100 g pouches. Regarding the NO precursors, the CM used was L-citrulline-DL-malate (2:1), sourced from HSN Store (Harrison Sport Nutrition S.L., Oviedo, Spain) (Sanitary Registration: 26.11001/GR). The NO_3_^−^ source was Oxystorm^®^, a high-content standardized extract of amaranth derived from Amaranthus hypochondriacus leaves with 9% NO_3_^−^ concentration, provided by Efficient Science (Black Fenix S.L., Cádiz, Spain) and distributed by EMFit (EMFit Nutrition S.L., Cádiz, Spain) (Sanitary Registration: 26.021287/CA).

The NIT + CM was formulated with the following ingredients: 60% strawberry purée, 22.15% apple purée, 10% grape concentrate, 4% CM, 3.65% Oxystorm^®^ (extract of amaranth, NO_3_^−^, and potassium), and 0.2% ascorbic acid. In contrast, the PLA contained 66.8% strawberry purée, 22% apple purée, 10% grape concentrate, 1% lemon concentrate, and 0.2% ascorbic acid.

Beetroot, a primary dietary source of NO_3_^−^, contains vitamin C (ascorbic acid) and phenolic compounds such as flavonoids, which help reduce oxidative stress and protect NO, enhancing its biological activity [54,55]. Amaranth, used as the NO_3_^−^ source in this study, naturally contains the antioxidant amaranthine [56]. To strengthen the antioxidant capacity and provide additional protection for NO against oxidative degradation, ascorbic acid was included to improve its stability and bioavailability.

Both the NIT + CM and PLA were designed to be nutritionally equivalent. Each 100 g pouch provided 207 kJ (49 kcal), 0.3 g of total fat (0.03 g saturated), 10.2 g of carbohydrates (8.2 g sugars), 1.5 g of fiber, 0.6 g of protein, 0.001 g of salt, and 21.5 mg of vitamin C.

Participants consumed two pouches daily, resulting in a total intake of 200 g per day, which provided 500 mg of NO_3_^−^ (8 mmol) and 8 g of CM.

To prevent any attenuation in the reduction of NO_3_^−^ to NO_2_^−^ by commensal bacteria in the oral cavity during the intervention, participants were instructed to refrain from using antibacterial mouthwash or toothpaste [21] and to ensure sufficient time elapsed between brushing their teeth and consuming the supplement. Additionally, to ensure the accuracy of the study results, participants were asked to refrain from using other ergogenic aids.

### 2.4. Blood Collection and Plasma Nitrate/Nitrite Analysis

Venous blood samples were collected from the antecubital vein using K_3_EDTA blood collection tubes (3 mL; BD Vacutainer^®^, Becton Dickinson, Franklin Lakes, NJ, USA). Samples were immediately centrifuged at 1000× *g* for 5 min at 4 °C. Plasma aliquots were snap-frozen in dry ice and stored at −80 °C until analysis at an external laboratory (Laboratorios MUNUERA S.L., Murcia, Spain).

Before analysis, plasma samples were thawed at room temperature and deproteinized using 10 kDa molecular weight cut-off filters (ultrafiltration), as recommended by the assay manufacturer. Total plasma NO_3_^−^ and NO_2_^−^ concentrations were quantified using a commercial colorimetric assay kit (Nitrite/Nitrate Assay Kit, Bioquochem^®^, KB03010; Oviedo, Spain), based on the Griess reaction. This colorimetric method has been widely used in plasma for the determination of NOx species [57].

The method involves enzymatic reduction of NO_3_^−^ to NO_2_^−^, followed by reaction with Griess reagents to form an azo dye measurable at 540 nm. Native NO_2_^−^ was quantified in parallel by omitting the enzymatic reduction step. NO_3_^−^ concentration was calculated by subtracting native NO_2_^−^ from total NO_2_^−^ values.

All samples and standards were assayed in duplicate, and standard calibration curves (0–200 µM) were generated using sodium nitrite solutions, following the manufacturer’s instructions. Absorbance was measured at 540 nm using a microplate reader.

### 2.5. Training and Match Data

External load monitoring in team sports is commonly performed using GPS devices [58]. For this study, a lower-limb wearable GPS tracker (calf-worn) developed by Oliver Sports (Sport Data Innovation SL, Barcelona, Spain) was utilized to monitor player performance metrics during pre-season training sessions and matches. The device integrates inertial measurement unit (IMU) sensors with a sampling rate of 50 Hz and GPS technology with a sampling rate of 25 Hz, enabling precise measurement of variables such as time, maximal speed, total distance covered (TD), and distance per minute.

This system is currently included in the FIFA Innovation Programme [59], which aims to explore emerging electronic performance and tracking systems (EPTS) [60] to improve data accuracy and player performance insights.

The wearable tracker incorporates AI-supported software that provides actionable insights for players and coaching staff, enhancing individual and team performance. The software automatically categorized movement into four predefined speed thresholds: walking (0 to 7.1 km/h), jogging (7.1 to 14 km/h), high-intensity running (HIR) (14 to 22.3 km/h), and maximal-intensity running (MIR) (22.3 to 36 km/h). These thresholds were fixed by the system and could not be adjusted. GPS data from training sessions and matches during pre-season were obtained from the tracking software and exported into Excel for further processing and analysis. Goalkeepers did not wear GPS devices and were therefore excluded from this part of the study.

### 2.6. Countermovement Jump and Wingate Anaerobic Tests

Explosive strength, anaerobic power, and neuromuscular fatigue in the lower limbs were measured and analyzed using the CMJ test [61,62,63], while anaerobic power and anaerobic capacity were analyzed using the WAnT [64]. The methodology, equipment, and variables used are detailed in a previous publication [65].

### 2.7. Statistical Analyses

SPSS Statistics 27 (SPSS, Inc., Chicago, IL, USA) was used for statistical analysis. Descriptive statistics were calculated and used to describe the GPS-derived training and match data, physical performance variables of CMJ and WAnT, and plasma NO_2_^−^ and NO_3_^−^ concentration.

The statistics were reported as mean ± standard deviation (SD). The Shapiro–Wilk and Levene tests were used to verify the normality and homoscedasticity of the data, respectively. Depending on the distribution of the variables, parametric (Student’s *t*-test) or non-parametric (Mann–Whitney U test) analyses were applied.

Additionally, a two-way independent measures ANOVA was performed to examine the main effects of week (Week 1 to Week 4) and group (NIT + CM vs. placebo), as well as their interaction on performance and physiological outcomes. In case of non-normally distributed dependent variables, the Kruskal–Wallis test was used as a non-parametric alternative. Bonferroni post hoc tests were applied if the homogeneity of variance assumption was met; otherwise, Games–Howell post hoc tests were used. In case of a significant interaction, simple main effects were explored using Bonferroni correction to account for multiple comparisons. Effect sizes were estimated using partial eta squared (η^2^p) and classified as small (0.01 ≤ η^2^p < 0.06), medium (0.06 ≤ η^2^p < 0.14), or large (η^2^p ≥ 0.14) [66]. The level of statistical significance was set at *p* ≤ 0.05.

## 3. Results and Discussion

A total of 42 professional female soccer players were enrolled in the study. All participants met the inclusion criteria and agreed to participate, so no exclusions were necessary. Players were randomly assigned to two equal groups (NIT + CM and placebo, *n* = 21 each). During the intervention period, five players in the NIT + CM group and three in the placebo group discontinued participation. In total, 34 players completed the study and were included in the final analysis (NIT + CM: *n* = 16; PLA: *n* = 18). A flow chart illustrating participant progression is presented in Figure 1.

The demographic characteristics of the players are displayed in Table 1. The anthropometric measurements and body composition of the players are available in detail in a previous publication by our research team [67].

To the best of our knowledge, this is the first study to assess the chronic effects of combined CM and NO_3_^−^ supplementation in professional soccer players. As this combination has been largely unexplored in the literature, investigating its potential effects under real-world training conditions may provide relevant insight for performance and recovery strategies.

GPS-derived data from training sessions were used to assess player performance, evaluate changes throughout pre-season, and explore potential differences between the NIT + CM supplementation group and the placebo group. The interaction between the type of product (NIT + CM vs. PLA) and time (the different weeks of the study) was not statistically significant for any of the variables analyzed. This indicates that the changes observed over the course of the weeks were similar in both groups, with no differences attributable to the product administered. Table 2 summarizes ANOVA results, while Table 3 presents weekly descriptive values by group and the entire sample.

Vmax showed significant main effects for both product and week. The product factor had a large effect size (*p* < 0.001; η^2^p = 0.351), with post hoc analysis confirming that values were consistently higher in the NIT + CM group compared to placebo throughout pre-season period. The week factor also had a large effect size (*p* < 0.001; η^2^p = 0.160), although smaller compared to the product factor. Post hoc tests for the week factor indicated progressive improvements from week 1 to weeks 3 and 4, and from week 2 to weeks 3 and 4 (all *p* < 0.001), peaking in week 4. Weekly comparisons using independent *t*-tests revealed that the NIT + CM group consistently reached higher maximum speeds than the placebo group throughout the pre-season period (all *p* < 0.001), indicating a stable between-group difference across the measured weeks.

Although previous studies on the chronic effects of NO precursor supplementation on Vmax in elite female soccer players are limited, some findings from team-sport athletes in sprint-related protocols provide indirect support. Improvements in cycling sprint performance have been reported following 5 to 7 days of beetroot juice supplementation (8.2–12.8 mmol NO_3_^−^/day), with increases in mean power [68] and total work [69]. Enhanced sprint times have also been observed in running-based tests under similar protocols [70]. These data suggest a potential benefit of NO_3_^−^ for explosive efforts, particularly following repeated-day administration [71].

From a physiological perspective, these effects may be mediated by the increased NO availability. Elevated NO levels have been associated with enhanced calcium handling, improved excitation–contraction coupling, and increased recruitment of type II muscle fibers [9,16], all of which are closely involved in generating Vmax in soccer [72,73]. High-speed running demands rapid activation of the posterior chain and strong neuromuscular coordination under fatigue [74,75], which could benefit from improved contractile efficiency and reduced ATP cost. Furthermore, the NO-induced increases in muscle perfusion and mitochondrial efficiency [76] could support the maintenance of sprinting capacity during consecutive sessions, particularly under high training loads. The malate component may further contribute by enhancing the rate of ATP production during high-intensity efforts [35].

Although our intervention combined NO_3_^−^ and CM, previous evidence suggests that CM alone may contribute to sprint performance. In a study, three days of CM supplementation (8 g/day) improved repeated sprint performance in male team sport athletes, with faster sprint times and reduced fatigue [77].

Taken together, the consistently higher Vmax values in the NIT + CM group, along with a larger effect size for the product than for the training week, may reflect a beneficial neuromuscular adaptation in response to NO precursor supplementation. Nevertheless, given the observational nature of the study and the multifactorial determinants of sprint performance, these findings should be interpreted with caution.

Regarding baseline performance, the mean Vmax recorded during the pre-season training period was 26.79 ± 4.45 km/h. Although direct comparisons must be interpreted cautiously due to methodological differences and seasonal context, it is worth noting that this value exceeds the average reported in Spanish professional female players during a four-week in-season training period (23.55 ± 1.30) [78]. This difference may be partially explained by the fact that pre-season training sessions are typically more demanding and focused on accumulating higher training loads compared to in-season microcycles, which prioritize recovery and match preparation [79].

As for distance covered per minute, this variable showed a significant main effect for product (*p* < 0.001), with a small effect size (η^2^p = 0.035). Post hoc analyses revealed consistently higher overall values in the NIT + CM group. The effect of the training weeks was close to statistical significance (*p* = 0.053) and showed a small effect size (η^2^p = 0.025). Post hoc analysis for the week factor showed a significant improvement from week 1 to week 3 (*p* = 0.009). Independent weekly *t*-tests revealed significantly higher values in the NIT + CM group only in weeks 1 (*p* < 0.001) and 3 (*p* = 0.029).

Considering all findings, it may indicate a modest but consistent advantage for the NIT + CM group in sustaining work rate during training. This effect is likely to involve mechanisms similar to those proposed for Vmax, such as enhanced muscle efficiency, improved blood flow, and increased ATP production through higher NO availability. Further studies under controlled conditions are needed to clarify the physiological relevance of this finding and its potential impact on training outcomes.

The overall mean distance per minute was 70.66 ± 22.72 m/min. Similar to the findings for Vmax, this value is also higher than those reported in Spanish professional players during the in-season period (60.12 ± 10.31) [78].

In addition, TD and walking distance showed significant main effects for the product. TD showed a moderate effect size (*p* < 0.001; η^2^p = 0.113), with the NIT + CM group presenting higher overall values compared to placebo. Additionally, a significant main effect was found for the week factor, although with a small effect size (*p* = 0.003; η^2^p = 0.029), with post hoc tests revealing significantly higher values in week 3 compared to week 2 (*p* = 0.006). Independent weekly *t*-tests confirmed consistently higher TD covered by the NIT + CM group compared to the placebo across all weeks (all *p* < 0.001).

Although a direct comparison cannot be made with distance covered during training, previous studies have reported increased TD in team-sport athletes during the Yo-Yo IR1 test following chronic NO_3_^−^ supplementation. These effects have been observed with protocols ranging from 2 to 6 days and doses between 4.1 and 12.9 mmol/day of NO_3_^−^ [70,80,81]. While those findings come from controlled performance tests and did not involve CM, the present results appear to follow a similar trend, suggesting a potential improvement in locomotor output during physically demanding periods such as pre-season.

The overall mean TD was 5281.13 ± 1272.78 m across the pre-season period. In line with previous findings for Vmax and distance per minute, this value was also higher than that reported in Spanish professional players from the same reference study (4883.5 ± 780.49) [78] but lower than the value reported in a study of Australian National League players during pre-season training (6646 ± 111) [79].

Walking distance showed a moderate product effect size (*p* < 0.001; η^2^p = 0.133), consistently favoring the NIT + CM group throughout pre-season. There was no significant week effect, yet independent weekly *t*-tests indicated significantly higher walking distances in the NIT + CM group at weeks 1, 2, and 4 (*p* < 0.001) and week 3 (*p* = 0.004).

Therefore, the consistent advantage observed in both total and walking distances among players in the NIT + CM group may reflect greater fatigue tolerance and an enhanced capacity to remain active for extended periods during training. Rather than indicating reduced intensity, the accumulation of walking distance may suggest a more effective active recovery process, allowing players to continue moving without the need for complete passive rest [82,83,84]. Considering the moderate effect sizes observed for the product factor and the consistent differences between groups, these findings support the hypothesis that NIT + CM supplementation could have contributed to a more favorable physiological response under sustained pre-season training demands. Nevertheless, these interpretations should be approached with caution, as the observational nature of the study precludes any inference of causality.

Physiologically, increased NO availability may have enhanced peripheral blood flow during training [85], improving oxygen and nutrient delivery while facilitating the clearance of metabolic byproducts [86]. Moreover, the malate component has been proposed to reduce lactate production by promoting aerobic utilization of pyruvate, especially under high blood flow conditions [35]. This combination may reflect a synergistic effect that supports sustained low-intensity activity and delays fatigue accumulation, potentially contributing to the greater total and walking distances observed in the NIT + CM group.

Moreover, jogging, HIR, and MIR distances exhibited significant main effects only for week, with no significant product effects. Specially, jogging distance presented a small effect size (*p* = 0.001; η^2^p = 0.034), with post hoc analysis showing fluctuations across weeks (higher in week 1 vs. week 2, *p* = 0.003; week 2 vs. week 3, *p* = 0.029). HIR distance displayed a moderate effect size (*p* < 0.001; η^2^p = 0.061), with a significant decrease from week 1 to week 2 (*p <* 0.001), followed by significant increases from week 2 to weeks 3 and 4 (both *p* ≤ 0.001). MIR distance showed a small effect size (*p* < 0.001; η^2^ = 0.059), with a progressive increase and significant increments from weeks 1 and 2 to weeks 3 and 4 (*p* ≤ 0.037).

Comparisons with previous studies are inherently limited by methodological differences, such as the velocity thresholds used in this analysis, which were defined by the GPS system and may not match those in other research. The variations observed in jogging, HIR, and MIR distances across the pre-season period likely reflect differences in the physical and tactical emphasis of each microcycle. Small effect sizes for jogging and MIR distances suggest minor week-to-week adjustments, while the moderate effect size for HIR may indicate a progressive increase in exposure to high-intensity efforts. These patterns are consistent with the natural variability of training content in professional soccer and the specific objectives set by the coaching staff each week [87].

Finally, training duration showed a small but statistically significant product effect (*p* = 0.001; η^2^p = 0.013). Despite reaching statistical significance, the small effect size limits its practical interpretation and calls for cautious analysis.

**Table 2 nutrients-17-02381-t002:** Two-way ANOVA used to examine the effect of the product (NIT + CM vs. PLA) and the week on GPS-derived performance variables during pre-season training.

	Product		Week		Product × Week	
Variable	*p*-Value	η^2^p	*p*-Value	η^2^p	*p*-Value	η^2^p
Time (min)	0.001 *	0.013	0.098	0.009	0.668	0.003
Vmax (km/h)	<0.001 *	0.351	<0.001 *	0.160	0.995	0.000
TD (m)	<0.001 *	0.113	0.003 *	0.029	0.694	0.003
Distance × minute	<0.001 *	0.035	0.053 *	0.025	0.789	0.002
Walking (m)	<0.001 *	0.133	0.902	0.003	0.480	0.005
Jogging (m)	0.194	0.007	0.001 *	0.034	0.763	0.002
HIR (m)	0.507	0.000	<0.001 *	0.061	0.515	0.005
MIR (m)	0.082	0.003	<0.001 *	0.059	0.645	0.004

Vmax: maximal speed; TD: total distance covered; Distance × minute: distance per minute; HIR: high-intensity running; MIR: maximal-intensity running. * Statistically significant (*p* < 0.05).

**Table 3 nutrients-17-02381-t003:** Descriptive statistics of GPS-derived training performance variables by group and week: mean and standard deviations (mean ± SD).

Variable	Group	Total	Week 1	Week 2	Week 3	Week 4
Time (min)	NIT + CM	81.71 ± 21.12	84.63 ± 20.67	82.00 ± 25.10	77.17 ± 19.41	82.64 ± 16.62
	Placebo	76.62 ± 19.99	79.22 ± 22.01	74.28 ± 24.54	75.92 ± 16.74	77.85 ± 13.17
	All players	78.82 ± 20.62	81.5 ± 21.52	77.77 ± 25.01	76.41 ± 17.75	80.05 ± 14.97
Vmax (km/h)	NIT + CM	29.63 ± 4.04	28.48 ± 3.21 *	28.19 ± 4.33 *	31.18 ± 4.06 *	31.27 ± 3.23 *
	Placebo	24.62 ± 3.40	23.39 ± 2.83	23.06 ± 3.50	26.11 ± 2.89	26.35 ± 2.87
	All players	26.79 ± 4.45	25.54 ± 3.91	25.38 ± 4.65	28.08 ± 4.19	28.61 ± 3.9
TD (m)	NIT + CM	5761.54 ± 1315.70	6001.50 ± 1580.04 *	5521.58 ± 1336.87 *	5961.62 ± 1175.81 *	5686.26 ± 1090.62 *
	Placebo	4915.52 ± 1109.51	4949.58 ± 1297.76	4594.22 ± 1022.62	5213.75 ± 983.84	4971.47 ± 1058.21
	All players	5281.13 ± 1272.78	5393.51 ± 1509.73	5014.03 ± 1259.41	5504.96 ± 1199.21	5293.35 ± 1265.53
Distance × minute	NIT + CM	75.26 ± 27.37	72.48 ± 15.20 *	74.06 ± 30.13	83.51 ± 30.92 *	72.19 ± 28.22
	Placebo	67.15 ± 17.68	64.02 ± 12.65	67.49 ± 21.58	71.47 ± 18.48	64.99 ± 14.45
	All players	70.66 ± 22.72	67.59 ± 14.35	70.46 ± 25.92	76.16 ± 24.68	68.29 ± 22.06
Walking (m)	NIT + CM	3735.42 ± 1068.13	3727.05 ± 1143.61 *	3840.14 ± 1125.00 *	3690.77 ± 1052.16 *	3642.09 ± 945.23 *
	Placebo	2994.69 ± 788.09	2901.70 ± 828.61	2957.38 ± 837.43	3149.77 ± 777.51	2963.85 ± 677.06
	All players	3314.80 ± 989.76	3250.01 ± 1052.04	3570.01 ± 1069.71	3604.43 ± 928.58	3274.97 ± 875.68
Jogging (m)	NIT + CM	1560.69 ± 780.93	1720.05 ± 700.95	1351.38 ± 826.37	1715.61 ± 863.55	1558.22 ± 658.11
	Placebo	1462.65 ± 614.45	1580.24 ± 632.83	1317.36 ± 657.53	1505.78 ± 620.94	1486.10 ± 490.20
	All players	1505.02 ± 692.27	1639.24 ± 662.9	1332.76 ± 736.33	1587.48 ± 728.67	1519.18 ± 571.71
HIR (m)	NIT + CM	435.17 ± 374.52	535.58 ± 372.81	313.75 ± 334.79	516.34 ± 434.05	434.07 ± 331.95
	Placebo	433.16 ± 331.86	448.10 ± 290.91	304.37 ± 256.09	527.87 ± 407.23	483.25 ± 320.32
	All players	434.03 ± 350.55	485.02 ± 329.21	308.62 ± 293.32	523.38 ± 415.99	460.69 ± 325.12
MIR (m)	NIT + CM	30.26 ± 58.93	18.83 ± 26.03	16.31 ± 37.62	38.90 ± 60.29	51.87 ± 89.14
	Placebo	25.02 ± 38.75	19.54 ± 23.16	15.11 ± 24.83	30.33 ± 44.01	38.27 ± 54.22
	All players	27.28 ± 48.52	19.24 ± 24.3	15.65 ± 31.17	33.67 ± 50.88	44.51 ± 72.32

Vmax: maximal speed; TD: total distance covered; Distance × minute: distance per minute; HIR: high-intensity running; MIR: maximal-intensity running. * Statistically significant (*p* < 0.05).

GPS-derived data from match play were analyzed to evaluate performance changes throughout pre-season and to identify potential differences between the NIT + CM supplementation group and the placebo group. Table 4 summarizes the ANOVA results, while Table 5 presents weekly descriptive values by group and for the entire sample.

Vmax showed a significant interaction among product and training week (*p* = 0.022; η^2^p = 0.079, moderate effect size). Post hoc analysis within the NIT + CM group revealed progressive increases from week 1 to weeks 3 and 4 (both *p* < 0.001), as well as from week 2 to weeks 3 (*p* = 0.006) and 4 (*p* = 0.028). Additionally, significant main effects were observed for both product (*p* < 0.001; η^2^p = 0.413, large effect size) and week (*p* < 0.001; η^2^p = 0.253, large effect size). Post hoc analysis for the product factor indicated consistently higher overall Vmax values in the NIT + CM group compared to placebo. For the week factor, post hoc tests showed clear and significant increases from week 1 to weeks 3 and 4 (both *p* < 0.001), and from week 2 to weeks 3 (*p* = 0.021) and 4 (*p* = 0.003). Independent weekly *t*-tests further confirmed significantly higher Vmax values for the NIT + CM group in weeks 1, 3, and 4 (*p* < 0.001) and week 2 (*p* = 0.012).

From a physiological perspective, the influence of both the product and the training week on this variable appears to be substantial, with the effect of the product being particularly pronounced. In addition, a significant interaction between product and training week suggests that differences between the NIT + CM and placebo groups evolved over time, potentially reflecting an enhanced physiological response in the supplemented group as pre-season progressed. This hypothesis was further supported by consistently higher Vmax values in the NIT + CM group across all weeks, as confirmed by independent *t*-tests. While causality cannot be established due to the observational design, the magnitude and stability of the group differences, along with the significant interaction, are compatible with a potential neuromuscular benefit of NIT + CM supplementation.

Due to the physiological mechanisms of NO_3_^−^ and CM discussed earlier, the product-related advantage observed here may reflect improved sprint capacity under competitive load. However, interpreting these findings requires consideration of the complex and variable nature of match demands. In elite women’s soccer, performance metrics such as Vmax are influenced by several factors including physical readiness, tactical role, opponent characteristics [88], and contextual factors such as match status or competition phase [89]. While these variables introduce a degree of variability, Vmax appears to be relatively stable. According to a study on match load variability in elite female players, this metric displays low inter-match variability (CV ≈ 4.5%) compared to other high-intensity measures [90], suggesting that changes are more likely to reflect genuine physiological adaptations rather than contextual fluctuations.

Moreover, the ability to express peak velocity in matches may be partially conditioned by the neuromuscular capacities developed during training. The consistent advantage observed in both training and match Vmax values among players in the NIT + CM group raises the hypothesis that enhanced high-speed performance during training could have facilitated a more robust translation of this ability into match contexts. Still, this interpretation must remain speculative and subject to the limitations inherent to non-experimental designs.

Descriptively, the overall mean Vmax observed during matches was 29.52 ± 4.26 km/h, which aligns with previous reports indicating that elite female players can reach peak speeds of approximately 30–32 km/h during match play [91,92]. In comparison, this value is slightly lower than the average Vmax reported in male professional players throughout the season (≈30.7 km/h), although most players tend to reach peak velocities between 32 and 33.9 km/h [93]. Finally, compared to the average reported in Spanish female players during a four-week match period in the competitive season (25.3 ± 0.3), the value observed in the present study appears notably higher [78]. Considering that all teams followed their usual training routines, this difference may reflect a potential influence of the supplementation strategy implemented.

Distance per minute showed a marginally significant interaction between product and match week (*p* = 0.052; η^2^p = 0.064, moderate effect size). Within-group analysis showed a significant increase from week 1 to week 4 in the NIT + CM group (*p* < 0.001). No significant main effect was found for the product factor; however, a significant main effect was observed for week (*p* < 0.001; η^2^p = 0.117, moderate effect size). Post hoc tests confirmed a clear improvement from week 1 to week 4 (*p* < 0.001). Independent *t*-tests revealed significantly higher values in the NIT + CM group at week 4 (*p* = 0.004).

The moderate main effect of training week on distance per minute, with significant increases from week 1 to week 4 across all players, suggests that the final pre-season match was played at a higher overall intensity. Although no main effect was found for the product factor, the presence of an almost significant interaction between product and match week, combined with the descriptively progressive increase observed in the NIT + CM group, may point to a more favorable adjustment in this group over time. The fact that this culminated in significantly higher values for the supplemented group in week 4 may reflect a more favorable adjustment to competitive demands. This pattern aligns with the physiological mechanisms previously discussed and warrants further exploration in controlled settings.

Descriptively, the overall mean distance per minute observed during matches was 93.30 ± 44.39 m/min, which is slightly lower but comparable to the average reported in Spanish professional female players during a four-week in-season period (96.3 ± 8.8) [78]. TD exhibited no significant effects for the product factor or the Product × Week interaction. However, a significant main effect for week was detected (*p* = 0.033; η^2^p = 0.079, moderate effect size), with increased values from week 1 to week 4 (*p* = 0.007). Moreover, the overall mean TD covered during matches was 7545.4 ± 4653.78 m, which falls slightly below the average reported in a recent systematic review of professional female players, which indicated typical match demands ranging from 8800 to 10,800 m, with an observed range between 7800 and 11,200 m depending on competition level, position, and measurement protocols [94,95].

Although TD increased from week 1 to week 4, this pattern should be interpreted with caution. As seen with distance per minute, the last match of pre-season appears to have been the most intense, which may partially explain the observed increase. TD is also highly influenced by contextual factors such as tactical role, match dynamics, and playing position [88], which are known to affect locomotor output during competition. The high standard deviation in this sample further highlights the large inter-individual and match-to-match variability, suggesting that the weekly differences may be more reflective of situational demands than systematic adaptations or product-related effects.

Regarding the distances covered at different velocity thresholds, no significant effects were found for the product factor or the Product × Week interaction in any of the four categories. However, independent *t*-tests revealed significantly higher walking distance in the NIT + CM group during week 4 (*p* = 0.026), which may reflect the elevated overall match intensity observed that week rather than a direct effect of the supplementation. Jogging distance displayed a significant main effect for week (*p* = 0.011; η^2^p = 0.095, moderate effect size), with post hoc tests revealing differences between week 1 and all subsequent weeks. Similarly, HIR showed a significant week effect (*p* = 0.002; η^2^p = 0.143, large effect size), again driven by differences between the first and later matches. MIR also followed this pattern, with a significant week effect (*p* = 0.021; η^2^p = 0.103, moderate effect size) and a trend toward higher values in week 3 compared to week 1 (*p* = 0.056).

Overall, the changes observed across matches suggest lower physical demands in the first game, followed by a relatively stable load in the remaining weeks. One isolated difference was found in HIR during week 1, with higher values in the placebo group (*p* = 0.006). However, the absence of consistent group effects and the high variability across players and contexts suggest that match-related situational factors were the main drivers of these fluctuations.

**Table 4 nutrients-17-02381-t004:** Two-way ANOVA used to examine the effect of the product (NIT + CM vs. PLA) and the week on GPS-derived performance variables during pre-season matches.

	Product		Week		Product × Week	
Variable	*p*-Value	η^2^p	*p*-Value	η^2^p	*p*-Value	η^2^p
Time (min)	0.749	0.005	0.742	0.006	0.921	0.004
Vmax (km/h)	<0.001 *	0.413	<0.001 *	0.253	0.022 *	0.079
TD (m)	0.325	0.023	0.033 *	0.079	0.238	0.036
Distance × minute	0.160	0.002	<0.001 *	0.117	0.052 *	0.064
Walking (m)	0.134	0.036	0.184	0.042	0.420	0.024
Jogging (m)	0.541	0.005	0.011 *	0.095	0.279	0.032
HIR (m)	0.081	0.000	0.002 *	0.143	0.277	0.033
MIR (m)	0.962	0.018	0.021 *	0.103	0.501	0.020

Vmax: maximal speed; TD: total distance covered; Distance × minute: distance per minute; HIR: high-intensity running; MIR: maximal-intensity running. * Statistically significant at (*p* < 0.05).

**Table 5 nutrients-17-02381-t005:** Descriptive statistics of GPS-derived match performance variables by group and week: mean and standard deviations (mean ±SD).

Variable	Group	Total	Week 1	Week 2	Week 3	Week 4
Time (min)	NIT + CM	82.15 ± 31.30	80.73 ± 32.47	87.04 ± 32.16	76.21 ± 33.65	84.19 ± 29.48
	Placebo	77.75 ± 30.29	75.20 ± 30.75	78.70 ± 31.38	78.45 ± 26.07	78.61 ± 35.99
	All players	79.70 ± 30.70	77.69 ± 31.13	82.34 ± 31.49	77.54 ± 28.88	81.3 ± 32.55
Vmax (km/h)	NIT + CM	32.28 ± 4.26	29.22 ± 2.50 *	30.65 ± 4.40 *	35.03 ± 4.29 *	34.42 ± 2.67 *
	Placebo	27.33 ± 2.73	25.69 ± 2.34	27.32 ± 2.60	27.44 ± 2.81	29.04 ± 2.27
	All players	29.52 ± 4.26	27.29 ± 2.97	28.78 ± 3.83	30.52 ± 5.11	31.64 ± 3.66
TD (m)	NIT + CM	8296.85 ± 6216.69	5361.70 ± 3157.51	7895.66 ± 6448.85	8709.23 ± 7525.30	11,250.29 ± 6139.06
	Placebo	6946.41 ± 2768.35	5985.29 ± 1852.95	7572.10 ± 2991.31	6879.60 ± 3240.47	7369.49 ± 2666.64
	All players	7545.4 ± 4653.78	5703.67 ± 2500.19	7713.65 ± 4730.07	7622.89 ± 5371.41	9242.98 ± 4994.85
Distance × minute	NIT + CM	95.54 ± 62.48	64.63 ± 35.04	84.69 ± 59.72	98.79 ± 74.74	134.25 ± 58.98 *
	Placebo	91.51 ± 21.37	83.49 ± 14.72	97.39 ± 11.01	87.60 ± 31.49	98.52 ± 18.75
	All players	93.30 ± 44.39	74.97 ± 27.18	91.83 ± 40.04	92.14 ± 52.62	115.79 ± 46.06
Walking (m)	NIT + CM	4864.74 ± 3825.86	3780.37 ± 2622.78	4536.51 ± 4256.96	4568.32 ± 4380.76	6552.60 ± 3680.61 *
	Placebo	3768.59 ± 1791.96	3609.23 ± 1628.64	3896.78 ± 1799.28	3592.53 ± 1939.08	4018.39 ± 1907.63
	All players	4254.79 ± 2915.52	3686.52 ± 2098.34	4176.66 ± 3078.76	3988.95 ± 3138.33	5241.8 ± 3125.71
Jogging (m)	NIT + CM	2676.28 ± 2523.16	1282.38 ± 1091.75	2734.77 ± 2654.58	3169.10 ± 3001.26	3554.06 ± 2592.83
	Placebo	2442.78 ± 1248.90	1821.92 ± 545.27	2957.51 ± 1459.65	2523.50 ± 1425.19	2426.49 ± 1104.39
	All players	2546.35 ± 1915.95	1578.26 ± 865.78	2860.06 ± 2033.74	2785.78 ± 2184.02	2970.83 ± 2014.93
HIR (m)	NIT + CM	677.85 ± 680.38	271.70 ± 203.65	574.29 ± 401.43	856.78 ± 843.89	1021.42 ± 845.33
	Placebo	680.40 ± 372.76	510.05 ± 235.67 *	691.76 ± 294.60	685.19 ± 386.85	853.78 ± 494.13
	All players	679.27 ± 529.0	402.41 ± 249.26	640.37 ± 344.49	754.9 ± 608.19	934.71 ± 679.06
MIR (m)	NIT + CM	77.98 ± 124.37	27.25 ± 31.02	50.09 ± 51.07	115.04 ± 155.63	122.20 ± 176.75
	Placebo	54.64 ± 59.35	44.09 ± 38.02	26.05 ± 25.68	78.37 ± 84.35	70.84 ± 57.22
	All players	64.99 ± 94.2	36.48 ± 35.5	36.57 ± 40.03	93.27 ± 117.65	95.64 ± 129.71

Vmax: maximal speed; TD: total distance covered; Distance × minute: distance per minute; HIR: high-intensity running; MIR: maximal-intensity running. * Statistically significant at (*p* < 0.05).

To assess post-match recovery, performance outcomes from the CMJ test and WAnT conducted on the first day of pre-season were compared with those obtained the day after a pre-season match. This second assessment took place after a four-week supplementation period, with the final dose consumed on match day. The aim was to determine which group showed better maintenance of performance—or, conversely, a smaller decline—under conditions of accumulated fatigue.

The CMJ pre-WAnT was used to assess anaerobic power and explosive strength under post-match conditions (Table 6). None of the variables showed statistically significant differences between groups. However, the NIT + CM group showed slightly more stable values in jump height, CMJ_Pmean_ and CMJ_Ppeak_, while the placebo group presented a small decline in both power metrics.

Although the differences were small and not significant, this pattern may suggest that the NIT + CM group recovered slightly better in terms of anaerobic performance.

The CMJ post-WAnT was used to evaluate neuromuscular fatigue and recovery capacity the day after the match. None of the variables showed statistically significant differences between groups (Table 7).

Jump height increased slightly in both groups, with very similar values in the change from baseline to the final test. This suggests that explosive strength was not markedly affected by the intervention in either condition.

In contrast, CMJ_Pmean_ and CMJ_Ppeak_ values showed a slightly more favorable evolution in the NIT + CM group. Both groups improved from baseline, but the NIT + CM group showed a greater increase in power compared to the placebo group. Although these differences were not significant, the trend may point toward a better recovery of neuromuscular performance following high-intensity effort in the NIT + CM group.

As with the CMJ pre-WAnT, these results do not confirm a clear effect of the product but suggest a consistent pattern that warrants further exploration under similar post-match conditions.

The WAnT was used to evaluate anaerobic power and capacity under post-match conditions (Table 8). Several variables showed significant differences between groups, indicating a better preservation of performance in the NIT + CM group. P_mean_ and P_min_ decreased in both groups, but the reduction was significantly smaller in the NIT + CM group. The decline in power across the 30 s effort was also consistently lower in the NIT + CM group, with significant differences in both the first and second half of the test.

Although P_peak_ and FI (%) did not reach statistical significance, both followed the same pattern observed in other variables. TP_peak_ showed no relevant differences between groups.

These findings suggest that the NIT + CM product may have contributed to preserving anaerobic capacity under conditions of accumulated fatigue. In the context of female soccer, this finding becomes especially relevant, as a systematic review and meta-analysis [96] has shown that high-intensity anaerobic performance, including sprint capacity, remains impaired for up to 48 h following a match. Supporting the ability to maintain anaerobic output the day after competition may therefore have practical implications for performance management during congested schedules or periods of high training load.

To complement the performance data, plasma concentrations of NO_3_^−^ and NO_2_^−^ were analyzed at baseline and again the day after the match (Table 9). The NIT + CM group showed significant increases from baseline in both NO_3_^−^ (*p* < 0.001) and NO_2_^−^ (*p* = 0.008). At the final time point, NO_3_^−^ was significantly higher in the NIT + CM group compared to placebo (*p* < 0.001), while no significant between-group difference was observed for NO_2_^−^.

Importantly, plasma NO_3_^−^ concentrations in the NIT + CM group remained elevated the day after the final dose. This aligns with previous research showing that chronic supplementation leads to progressive increases in plasma NO_3_^−^, and that concentrations remain above baseline even 24 h after the last intake. For instance, in a study with 19.5 mmol/day over 8 days, values rose from 37 ± 15 μM to 270 ± 182 μM one day after the final dose, indicating prolonged systemic exposure [97]. Although absolute values are not directly comparable due to population and dose differences, this finding is consistent with the present observations using 8 mmol/day over four weeks.

Moreover, previous studies have shown that multi-day NO_3_^−^ intake results in higher plasma NO_3_^−^ compared to single bolus ingestion [45], and that NO_2_^−^ responses are more closely associated with total NO_3_^−^ intake across the protocol than with daily intake alone [12]. In the present analysis, NO_2_^−^ was the only biomarker that did not show a significant between-group difference at the final measurement, despite a significant increase from baseline in the NIT + CM group. This finding may be related to the markedly shorter plasma half-life of NO_2_^−^ compared to NO_3_^−^ [98], which limits its persistence in circulation during the recovery phase. In addition, NO_2_^−^ may be buffered or stored within skeletal muscle tissue [45], potentially reducing its detectability in plasma while still contributing to local NO availability. These mechanisms could help explain the absence of a significant difference between groups, even under conditions of elevated systemic NO_3_^−^ availability.

## 4. Limitations

This study has some limitations that should be acknowledged to contextualize the findings. The sample size, while adequate for a pilot investigation in elite athletes, may limit the generalizability of the results to female soccer. However, it is important to note that access to professional soccer teams is inherently difficult, and these limitations are common in studies involving elite athletes. Although certain external variables were controlled, such as the use of ergogenic aids and mouthwash, detailed monitoring of dietary intake was not performed, which could have influenced NO_3_^−^ availability. The study was conducted exclusively during the pre-season, a period with specific training demands, so caution is needed when extrapolating to other phases of the season. Additionally, while relevant performance and recovery metrics were included, no biochemical markers of muscle damage were assessed, due to limitations from the teams. The sample included players from two different professional teams, which may have introduced variability in training methods, coaching decisions, and recovery protocols. This reflects a broader limitation of studies in team sports, as standardization is often constrained by factors that lie outside the control of the research team. Finally, although the supplemented group showed consistent positive trends, not all differences reached statistical significance, suggesting that further research is warranted to confirm these preliminary findings.

## 5. Conclusions

This study evaluated the effects of chronic supplementation with NO_3_^−^ and CM during pre-season in professional female soccer players. Players who received the NIT + CM product consistently achieved higher Vmax values during both training sessions and matches, along with greater total and walking distances and a higher distance covered per minute in training. These results suggest a greater tolerance to high training loads during pre-season. Most importantly, the NIT + CM group showed better preservation of anaerobic capacity the day after a competitive match, with significantly smaller reductions in mean and minimum power during the WAnT. These findings were supported by elevated plasma NO_3_^−^ concentrations measured 24 h after the final dose, indicating sustained systemic exposure during the recovery phase. Overall, these results highlight the potential utility of NIT + CM supplementation to support neuromuscular performance and attenuate post-match fatigue during pre-season, a period characterized by high physical demands and limited recovery windows. This strategy may offer practical benefits for nutritionists aiming to optimize player readiness and recovery during pre-season, as well as to support performance across other high-demand phases of the competitive season.

## Figures and Tables

**Figure 1 nutrients-17-02381-f001:**
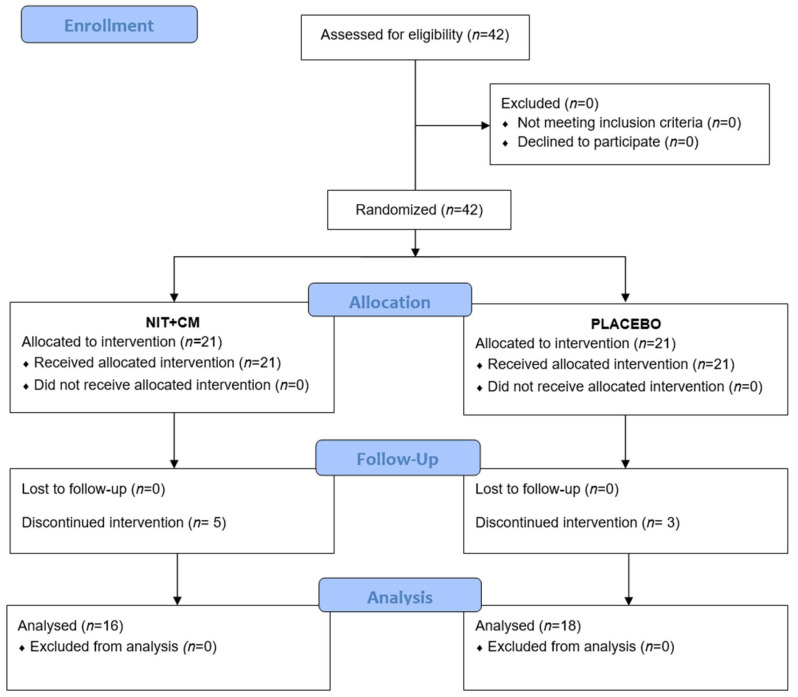
Flow diagram.

**Table 1 nutrients-17-02381-t001:** Demographic data of the players: mean and standard deviations (mean ± SD).

	Mean ± SD
*n*	34
Age	23.06 ± 4.29
Height (cm)	164.15 ± 5.84
Weight (kg)	58.39 ± 6.62
BMI (kg/m^2^)	21.64 ± 21.59
∑6 Skinfolds (mm)	74.57 ± 18.48

BMI: body mass index; ∑6 Skinfolds (triceps, subscapular, supraspinal, abdominal, front thigh, and medial calf).

**Table 6 nutrients-17-02381-t006:** Descriptive statistics at baseline and at the end of pre-season, and variation of CMJ before WAnT: mean and standard deviations (mean ± SD).

Variable	Group	Baseline	Final	∆	*p*-Value
				Final − Baseline	
Height_1_	NIT + CM	24.93 ± 4.30	25.57 ± 3.85	0.64 ± 0.91	0.437
	Placebo	26.33 ± 4.77	26.54 ± 4.90	0.21 ± 1.54	
CMJ_Pmean1_	NIT + CM	648.00 ± 66.36	654.20 ± 72.70	6.20 ± 24.99	0.424
	Placebo	673.43 ± 100.60	670.16 ± 91.56	−3.27 ± 30.67	
CMJ_Ppeak1_	NIT + CM	659.07 ± 74.00	661.67 ± 74.74	2.60 ± 26.48	0.478
	Placebo	708.20 ± 109.47	703.31 ± 108.06	−4.89 ± 24.31	

Height: jump height (cm); P_mean_: mean power (W); P_peak_: power peak (W).

**Table 7 nutrients-17-02381-t007:** Descriptive statistics at baseline and at the end of pre-season, and variation of CMJ after WAnT: mean and standard deviations (mean ± SD).

Variable	Product	Baseline	Final	∆	*p*-Value
				Final − Baseline	
Height_2_	NIT + CM	22.76 ± 2.78	23.70 ± 2.77	0.95 ± 1.06	0.931
	Placebo	24.49 ± 4.39	25.55 ± 4.65	1.06 ± 1.06	
CMJ_Pmean2_	NIT + CM	614.15 ± 56.17	632.04 ± 62.62	17.89 ± 21.65	0.563
	Placebo	664.81 ± 99.19	677.43 ± 83.02	12.62 ± 22.32	
CMJ_Ppeak2_	NIT + CM	623.99 ± 57.90	643.64 ± 62.76	19.65 ± 18.77	0.285
	Placebo	679.66 ± 103.49	689.49 ± 85.50	9.83 ± 22.99	

Height: jump height (cm); P_mean_: mean power (W); P_peak_: power peak (W).

**Table 8 nutrients-17-02381-t008:** Descriptive statistics at baseline and at the end of pre-season, and variation of WAnT: mean and standard deviations (mean ± SD).

Variable	Group	Baseline	Final	∆	*p*-Value
				Final − Baseline	
P_peak_	NIT + CM	635.09 ± 89.93	625.91 ± 81.80	−9.18 ± 25.30	0.070
	Placebo	631.71 ± 113.53	596.64 ± 108.63	−35.07 ± 39.14	
TP_peak_	NIT + CM	6.36 ± 1.21	5.91 ± 1.81	−0.46 ± 1.64	0.469
	Placebo	6.64 ± 2.27	5.57 ± 2.62	−1.07 ± 2.37	
P_min_	NIT + CM	378.82 ± 43.49	349.73 ± 45.33	−29.09 ± 28.04	0.036 *
	Placebo	404.79 ± 48.21	345.21 ± 24.79	−59.57 ± 37.95	
FI (%)	NIT + CM	39.70 ± 8.05	42.12 ± 7.95	2.49 ± 2.58	0.087
	Placebo	38.01 ± 6.02	42.89 ± 9.17	4.88 ± 3.95	
P_mean_	NIT + CM	517.94 ± 58.80	497.49 ± 51.74	−20.46 ± 21.06	0.004 *
	Placebo	520.27 ± 73.52	468.31 ± 57.52	−51.96 ± 23.38	
Split_0–10S_	NIT + CM	591.53 ± 88.83	585.95 ± 78.96	−5.58 ± 23.40	0.046 *
	Placebo	586.14 ± 111.71	553.59 ± 97.79	−32.55 ± 36.23	
Split_10–20S_	NIT + CM	541.87 ± 57.09	511.13 ± 55.72	−30.73 ± 33.16	0.019 *
	Placebo	543.39 ± 71.94	483.30 ± 65.55	−60.09 ± 23.44	
Split_20–30S_	NIT + CM	431.28 ± 40.27	398.87 ± 37.02	−32.41 ± 23.94	0.021 *
	Placebo	443.85 ± 51.27	386.36 ± 41.52	−57.48 ± 25.08	
Split_0–15S_	NIT + CM	583.25 ± 78.48	571.01 ± 70.53	−12.23 ± 23.64	0.012 *
	Placebo	579.49 ± 97.27	537.91 ± 85.21	−41.58 ± 28.16	
Split_15–30S_	NIT + CM	457.00 ± 43.76	425.35 ± 40.73	−31.65 ± 23.32	0.011 *
	Placebo	467.56 ± 55.74	408.97 ± 46.03	−58.59 ± 23.88	

P_peak_: peak power (W); TP_peak_: time to P_peak_ (s); P_min_: minimum power (W); FI (%): fatigue index; P_mean_: mean power (W); Split: P_mean_ calculated every 10 and 15 s. * Statistically significant at (*p* < 0.05).

**Table 9 nutrients-17-02381-t009:** Descriptive statistics at baseline and at the end of pre-season, and variation of NO_2_^−^ and NO_3_^−^ plasma concentration: mean and standard deviations (mean ± SD).

Variable	Group	Baseline	Final	∆	*p*-Value
				Final − Baseline	
NO_2_^−^	NIT + CM	4.43 ± 2.59	13.20 ± 14.35	8.78 ± 13.19 *	0.008 *
	Placebo	5.02 ± 2.33	7.12 ± 3.55	1.92 ± 2.50	
NO_3_^−^	NIT + CM	12.32 ± 4.20	86.10 ± 40.18 *	73.78 ± 40.74 *	<0.001 *
	Placebo	12.85 ± 5.66	26.56 ± 15.05	14.82 ± 14.41	

* Statistically significant at (*p* < 0.05). Concentrations were expressed in micromoles per liter (µmol/L).

## Data Availability

The data are contained within the article.

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
