# Peer review of "Impact of Chronic Nitrate and Citrulline Malate Supplementation on Performance and Recovery in Spanish Professional Female Soccer Players: A Randomized Controlled Trial"

_nutrients, 2025, doi:10.3390/nu17142381_

Round 1
Reviewer 1 Report
Comments and Suggestions for Authors
Introduction
The introduction is overall well-written, detailed, and logically structured. It provides a comprehensive background on nitric oxide production pathways and the physiological relevance of both nitrate and citrulline malate supplementation.
The authors clearly identify a gap in the literature namely, the lack of studies combining these two supplements in professional female soccer players and they justify the relevance of their research question.
A few sentences could be shortened to enhance conciseness, and the repetition of certain biochemical explanations (e.g., NO₃⁻-NO₂⁻-NO pathway) could be reduced. Additionally, it would strengthen the introduction to briefly mention studies with neutral or non-significant findings related to these supplements, to ensure a more balanced view of the literature.
Overall, the introduction effectively sets the stage for the research and supports the hypothesis with solid scientific rationale.
Methods
The methodology is overall rigorous, well-structured, and transparently described. The authors should be commended for implementing a randomized, double-blind, placebo-controlled trial, with team stratification, which strengthens the internal validity of the study.
The description of supplementation procedures, product composition, and dosage is highly detailed and allows for replication. It is appreciated that participants were instructed to avoid antibacterial mouthwash and other ergogenic aids that might interfere with NO bioavailability. The use of amaranth extract instead of beetroot adds novelty.
The GPS tracking procedures are clearly described, including device specifications, sampling frequency, and movement classification thresholds. However, the inability to adjust speed thresholds may limit the granularity of position-specific analyses. Exclusion of goalkeepers from GPS analysis is appropriate and justified.
The counter-movement jump (CMJ) and Wingate test protocols are referenced to previous work, which is acceptable, but a brief summary of test conditions (e.g., familiarization, time of day, recovery intervals) would improve methodological transparency.
The statistical plan is comprehensive and adheres to good practice. Effect sizes are reported and interpreted according to standard thresholds, which is essential in sport science research. The use of both parametric and non-parametric approaches depending on data characteristics is also a strength.
Minor suggestion: The authors may consider briefly justifying their sample size or reporting a priori power calculation, if applicable.
Results
The Results section presents a broad range of data and systematically addresses all pre-defined outcome measures, including GPS-derived performance metrics, countermovement jump (CMJ), Wingate anaerobic test (WAnT), and plasma nitrate/nitrite concentrations. The use of statistical analyses is appropriate, incorporating t-tests, ANOVA, and effect size calculations (η²p), which strengthen the credibility of the findings.
However, there are several weaknesses in terms of prioritization and interpretation. Although numerous variables are reported, it is not entirely clear which outcome is the primary result of the study. The performance differences between the NIT+CM and placebo groups are outlined, but the practical significance of these differences is not sufficiently emphasized. For example, the observed increase in maximal intensity running (MIR) in the intervention group appears to be meaningful, yet the authors do not elaborate on its real-world implications for training or competition. In contrast, the CMJ and WAnT results seem secondary and are presented without a clear connection to the study’s main hypotheses.
Additionally, there is no discussion of inter-individual variability, which could be particularly relevant in applied sport settings. The absence of individual response analysis may overlook potential responders or non-responders to the supplementation strategy. Furthermore, while the statistical significance is reported, the lack of accompanying confidence intervals makes it harder to assess the precision and robustness of the effects. Overall, while the data are comprehensive, the narrative would benefit from greater focus, clearer prioritization of key findings, and more context regarding their applied relevance.
Discussion
The discussion attempts to relate the findings to current knowledge regarding nitrate- and citrulline-based supplementation; however, much of the content reiterates the introduction rather than offering deeper critical insight. The interpretation of group differences in GPS and CMJ variables remains superficial and lacks a comprehensive explanation of the physiological mechanisms or alternative explanations for the observed outcomes.
The discussion does not contextualize the findings adequately within the specific demands of professional women’s soccer. Furthermore, the inclusion of exclusively female athletes is not addressed in terms of potential sex-specific responses, limiting the translational value of the findings. While the study hypothesized synergistic effects of NO₃⁻ and CM co-supplementation, the discussion does not sufficiently explain why this combination led to limited or partial effects. Key issues such as individual variability in supplement responsiveness, possible differences in habitual diet, fluctuations in training load during the intervention, or the adequacy of the 4-week duration are not critically discussed.
In addition, the lack of significant effects in the Wingate anaerobic test (WAnT)—an outcome closely related to the rationale for CM use—is not addressed. Given its relevance to repeated high-intensity efforts, this omission is a missed opportunity to enhance the interpretation of findings.
The limitations section is relatively brief and omits several critical points. Notably, it does not address potential differences in dietary intake, placebo effects, training variability, or GPS-derived data contextualization. Additionally, there is no mention of the study’s statistical power or a priori power calculation, which is particularly relevant considering the small sample size.
The discussion would benefit from a more balanced and nuanced interpretation of the data, improved integration of recent literature (including studies reporting null findings), and a clearer explanation of both the strengths and weaknesses of the experimental approach.
Reviewer 2 Report
Comments and Suggestions for Authors
The article is prepared very thoroughly, supported by scientific evidence. The methodology is very elaborate which demonstrates the competence of the authors. All works cited are included correctly. The purpose of the research introduces a new quality to science by capturing women in sport.
The article does not provide an explanation regarding the differences in training between the first and second women's leagues. It includes a reference to previous publications which introduces a slight misinformation when reading (225).
No explanation in the text regarding the study group regarding the reasons for the reduction of the study group during the intervention.
Reviewer 3 Report
Comments and Suggestions for Authors
This study sought to investigate nitrate supplementation on the performance of female soccer players. The study has a significant limitation. Participants from two different soccer teams were recruited and allocated to the supplemented or placebo group, then compared each training and match load without controlling for these variables during the study. Therefore, these training and match load data are not reliable for comparing the supplement's effect. I recommend that participants remove these results and focus on laboratory data.
Below is a list for more detailed review.
Introduction
-Write NO, NO₃⁻, CM and NO₂ full in the first appearance.
- Materials and Methods
2.3. Supplementation
-What was the adherence to supplementation? Was there control?
2.4. Blood Collection
-Was the blood collected while fasting? At what time? And after supplementation, how long after the last dose was the blood collected?
-Please provide data/reference about validity and reproducibility of GPS data.
2.5. Training and Match data
-The match data is very poorly described. What type of match data was included? Official matches or friendlies? Did the athletes play the entire match or incomplete matches?
- Were the players' positions distributed equally across the two supplemented groups?
-How was the training controlled if all athletes did not come from the same team?
Did all athletes receive the same stimulus for small-sided or full-field play? For the number of sprints, total distance? If these actions were not the same for both groups, comparisons cannot be made as presented in table 2.
-The comparisons cannot be made with data from training and matches because the participants do not belong to the same team; they are from the same positions? Did they play the same game? Did they have the same coach training?
2.5. (2.6????) Countermovement jump and Wingate anaerobic tests
-Need to describe the tests. What time was it? How many attempts per jump? What equipment was used?
-Please provide the CV and ICC of CMJ and WAnT laboratory tests.
Result
-Place table 2 near the first mention in line 279. This should be done for the other tables as well.
The literature review from 290 to 312 doesn't discuss the findings; it merely provides a review and adds nothing to the discussion. I recommend revising the discussion and removing the literature reviews that don't discuss the results.
